# Large Displacement Motion Transfer with Unsupervised Anytime Interpolation

**Guixiang Wang** [1]   **Jianjun Li** [2]

## Abstract

Motion transfer is to transfer pose in driving video to the object of the source image so that the object of the source image moves. Although great progress has been made recently in unsupervised motion transfer, many unsupervised methods still struggle to accurately model large displacement motions when large motion differences occur between source and driving images. To solve the problem, we propose an unsupervised anytime interpolation-based large displacement motion transfer method, which can generate a series of any time interpolated images between source and driving images. By decomposing large displacement motion into many small displacement motions, the difficulty of large displacement motion estimation is reduced. In the process, we design a selector to select optimal interpolated images from generated interpolated images for downstream tasks. Since there are no real images as labels in the interpolation process, we propose a bidirectional training strategy. Some constraints are added to the optimal interpolated image to generate a reasonable interpolated image. To encourage the network to create high-quality images, a pre-trained Vision Transformer model is used to design constraint losses. Finally, experiments show that compared with the large displacement motion between source and driving images, the small displacement motion between interpolated and driving images makes it easier to realize motion transfer. Compared with existing state-of-the-art methods, our method significantly improves motion-related metrics.

## 1. Introduction

In recent years, motion transfer task has become one of the research hotspots in computer vision because it has been widely used in film animation, game production, face exchange and other fields(Siarohin et al., 2019a; Oquab et al., 2021; Wang et al., 2021; Zakharov et al., 2020). In the work, we aim to animate an object in a still image by transferring the pose in the driving video to generate a video with the same pose as the driving video. To make the generated video more vivid, it is necessary not only to accurately transfer motion patterns in driving video but also to ensure that the identity information of the generated video frame and source image are consistent.

At present, motion transfer tasks can be divided into supervised and unsupervised methods. Supervised methods perform motion transfer by using prior knowledge of target object, such as landmarks(Chan et al., 2019; Ha et al., 2020; Ren et al., 2020; Siarohin et al., 2018; Zakharov et al., 2019) and 3D models(Doukas et al., 2021; Liu et al., 2019; Thies et al., 2020; Blanz & Vetter, 1999). However, these methods usually rely on pre-trained models to extract object-specific representations, and often fail to obtain satisfactory results for data objects that do not appear in training. Recently, researchers have proposed some unsupervised motion transfer methods that do not require prior knowledge. These methods use different transformation methods to model motion to improve the accuracy of motion transfer. For example, both FOMM(Siarohin et al., 2019b) and MRAA(Siarohin et al., 2021) methods used a local linear affine transformation to model motion. However, in the real world, the motion of objects is usually not locally linear, which makes it difficult for affine transformation to represent complex motions accurately. To this end, TPSMM(Zhao & Zhang, 2022) introduced a more flexible nonlinear transformation (Thin-plate splines) to approximate motions. However, the set of keypoints detected by TPSMM is messy, and accuracy is not high, which limits the motion representation ability of thin-plate spline transformation. To solve the problem, CPABMM(Wang et al., 2024) utilized continuous piecewise affine transformation to model motion. However, these motion models are limited in their ability to simulate finer motions and often produce artifacts around local regions. To this end, Tao et al.(Tao et al., 2023) proposed a motion refinement module to compensate for previous motion models,

---

[1]School of Computer Science and Engineering, Hangzhou Dianzi University, Zhejiang, China [2]School of Information Science and Technology, Hangzhou Normal University, Zhejiang, China. Correspondence to: Jianjun Li <lijjcan@gmail.com>.

*Proceedings of the 42nd International Conference on Machine Learning*, Vancouver, Canada. PMLR 267, 2025. Copyright 2025 by the author(s).

achieving finer motion modeling in local regions.

Although these methods focusing on accurate motion estimation have achieved good performance, it is often difficult to obtain accurate animation results when the motion difference between source and driving images is large. It is well known that if the motion difference between two images is smaller, then the pose of the driving image is more straightforward to transfer accurately. In the case of large displacement motions, previous methods attempt to model complex large displacement motions directly, which will be a serious challenge. To solve the problem of large displacement motion, CoP(Fu et al., 2023) proposed a method based on chain-of-pose. However, in the real world, when the source and driving images have different identity information, it is difficult to obtain a segment of the pose chain between the source and driving images. In some datasets, there may also be a large displacement motion between two adjacent frames in chain-of-pose. To solve the above problems, a novel large displacement motion transfer model based on unsupervised anytime interpolation is proposed.

Specifically, by inserting a series of intermediate images between the source and the driving images, complex large displacement motion is decomposed into many small displacement motions to improve the performance of the motion transfer model in the case of large displacement motion. As shown in Figure 1, we propose an unsupervised keypoint-based interpolation method to realize interpolation between source and driving images at any time. Firstly, the keypoints of source and driving images are obtained, and the keypoint information of interpolated images at different times is estimated according to the keypoints of the two images. Then, a dense motion network is used to predict motion flow from the source image to interpolated images at different times. An optimal interpolation selector is designed to obtain a better transfer effect for the downstream motion transfer task. The selector requires that the optimal interpolated image should meet the following two requirements: 1) maintain the integrity of identity information; 2) The motion between interpolated and driving images should be small. Since an optimal interpolated image is generated unsupervised, a bidirectional training strategy is proposed to complete the interpolation without paired data, and reasonable constraints are designed. Compared with large displacement motion estimation from source image to driving image, small displacement motion estimation from interpolated image to driving image will be more accurate. Finally, we design a structural consistency loss using a pre-trained Vision Transformer (ViT) model(Caron et al., 2021; Tumanyan et al., 2022) as an external semantic before extract structural information from both generated target image and driving image. The appearance consistency loss is designed by extracting appearance information from pairwise optimal interpolated images generated during the bidirectional training. These constraint losses encourage the network to generate images with good quality.

Our main contributions can be summarized as follows:

- To solve the problem of large displacement motion, we propose a keypoint-based anytime interpolation method, which can realize unsupervised interpolation between source and driving images at any time and decompose complex large displacement motion into many small displacement motions.

- A motion transfer method is proposed based on the optimal interpolated image. We design a selector for optimal interpolation, which requires that the optimal interpolated image not only preserves the identity information of the source image well but also has a small motion to the drive image. Compared with large displacement motion from the source image to the drive image, small displacement motion from the optimal interpolated image to the driving image is easier to estimate so that the generated target image can learn a more accurate pose.

- A bidirectional training strategy is proposed, which uses a pre-trained ViT model to construct constraint terms, such as structural consistency loss and appearance consistency loss, to reduce the solution space of interpolated images and improve the quality of generated images.

- The experiments and analysis are carried out on three datasets. Compared with other state-of-the-art motion transfer models, the proposed method can achieve the best performance in motion-related metrics.

## 2. Related Work

### 2.1. Supervised Motion Transfer Methods

Supervised motion transfer methods are designed to handle specific objects (e.g., faces, bodies, etc.) that exhibit strong regularity in shape and structure. The video can be generated more stably by introducing the corresponding structural prior as an intermediate representation. For example, HeadGAN(Doukas et al., 2021) used 3DMM(Blanz & Vetter, 1999) as intermediate representation and SPADEs(Park et al., 2019) to fuse identity features of source image, and finally renders 3D image back to animation frame. GANimation(Pumarola et al., 2018) used Facial Action Coding System(Ekman & Rosenberg, 1997) to describe facial expressions. With the help of these structural priors, supervised motion transfer methods can generate videos more accurately. However, these methods must rely on explicit structural representations, annotated or extracted by pre-

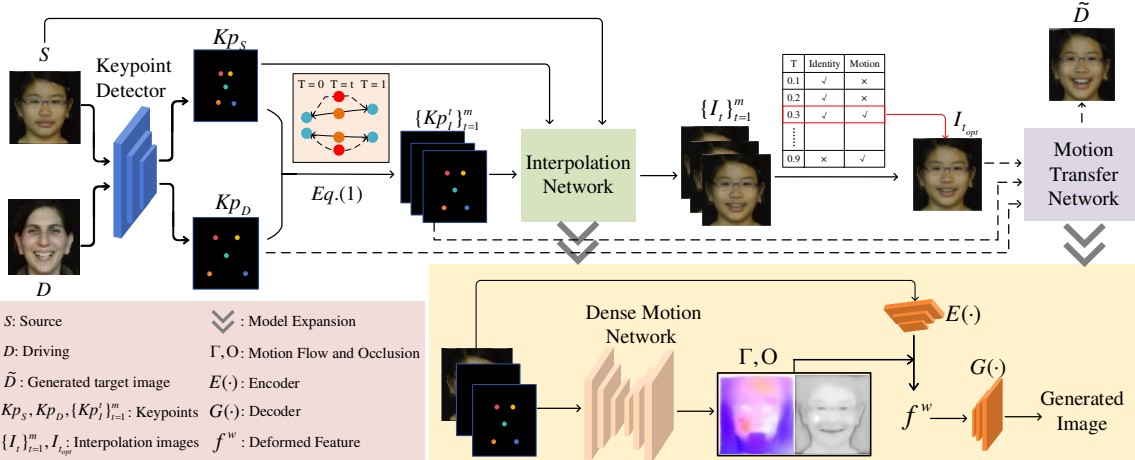

Figure 1. Overview of our model. The model consists of two modules. The first module is an arbitrary interpolation method based on keypoints. At Eq. (1), we visually describe the motion of keypoints of interpolated images. The blue circles represent the corresponding keypoints at time $T = 0$ (source image) and $T = 1$ (driving image), the red circles represent true keypoints at time $t$, and the orange circles represent keypoints of the interpolated image at time $t$ under linear motion assumption. After obtaining a series of anytime keypoints, we input into the interpolation network to generate the corresponding interpolated images. The second module is the motion transfer method based on the optimal interpolated image. To improve the motion transfer effect, we designed a selector to select an optimal interpolated image from a series of interpolated images for motion transfer. These two modules use the same network architecture with shared weights, where $f^w$ is the result of deformed features of the input image.

trained models, to perform motion transfer for specific objects. Due to the limitations of these structural priors, these supervised methods do not generalize well to new objects.

### 2.2. Unsupervised Motion Transfer Methods

In recent years, many unsupervised motion transfer methods have been proposed. Monkey-Net(Siarohin et al., 2019a) is a general object image animation framework, which consists of a keypoint detector, a motion prediction network, and an image generation network. Based on the assumption of local linear motion around each keypoint, FOMM(Siarohin et al., 2019b) used local affine transformation to model motion. Zhao et al.(Zhao et al., 2021) transferred motion from sparse landmarks to face images and combined global and local motion estimation into a unified model that can produce not only global motion but also subtle local motion. MRAA(Siarohin et al., 2021) used an inference algorithm based on PCA(Kurita, 2014) to calculate affine parameters, which improves stability and performs better for joint movements. TPSMM(Zhao & Zhang, 2022) predicted several sets of keypoints from source and driving images and used each set of keypoint sets to compute a thin-plate spline (TPS) transformation to increase the flexibility of the motion model. CPABMM(Wang et al., 2024) used the inference method of gradient descent to estimate transformation based on continuous piecewise affine (CPA), and to overcome the problem of chaotic keypoint prediction in TPSMM, a keypoint semantic loss based on SAM model is designed to

improve keypoint detector. Although these methods have achieved good transfer performance, they are often unsatisfactory in the case of large displacement motion. Therefore, we propose a large displacement motion transfer method based on unsupervised anytime interpolation to improve motion estimation accuracy by decomposing large displacement motion into multiple small displacement motions.

## 3. Methods

### 3.1. Motivation

Given a source image $S$ and a driving video $D = d_{1:n}$ with n frames, our goal is to have the object of the source image imitate motion in the driving video to generate a realistic video. However, when the pose difference between the source image and some frames of driving video is large, it is often difficult to accurately estimate the motion of two images, resulting in a high-quality video that cannot be generated. To solve this problem, we propose a motion transfer model based on unsupervised anytime interpolation. Our model consists of two modules: keypoint-based anytime interpolation and motion transfer based on the optimal interpolated image. First, we predict interpolated images for $m$ moments unsupervised for a given source image and driving images. A selector is then set to select the most suitable image from m interpolated images for the downstream motion transfer task. We use the optimal interpolated image to replace the source image and realize motion transfer from

the interpolated image to the driving image. We use the same network architecture for these two modules: a keypoint detection network, a dense motion network, and an image generation network. An optimal interpolated image and final target image are generated end-to-end. Compared with traditional motion transfer methods, the target image generated by the proposed method has a more accurate pose.

### 3.2. Large displacement motion transfer with unsupervised anytime interpolation

**Keypoint-based anytime interpolation method.** In FOMM(Siarohin et al., 2019b), it is assumed that motion consists of a series of linear transformations. Inspired by it, a keypoint-based anytime interpolation method is proposed. In Figure 1, we visually describe the motion of keypoints of interpolated image, where when $T = 0$, it can be represented as keypoints $(Kp_S)$ of source image, and when $T = 1$, it can be represented as keypoints $(Kp_D)$ of driving image. In the unsupervised case, prior motion information of each keypoint at time $t \in (0, 1)$ cannot be obtained. Therefore, we assume that the motion of each keypoint is linear, then anytime keypoints within $t \in (0, 1)$ can be expressed as follows:

$$Kp_I^t = (1 - t)Kp_S + tKp_D, t \in (0, 1) \qquad (1)$$

Taking the interpolated image at time $t$ as an example, here we apply the TPS transformation to estimate the motion from $S$ to $I_t$ such that $\Gamma(S) = I_t$. Like TPSMM(Zhao & Zhang, 2022), we get $K + 1$ transformations ($K$ TPS transformations and an affine transformation), where affine transformation is used to model the motion of the background. Then, we feed $S$, $Kp_S$, and $Kp_I^t$ into dense motion network to predict $K + 1$ contribution maps $\{M_k\}_{k=0}^{K}$, which are combined with $K + 1$ transformations to approximate the motion $\Gamma$:

$$\tilde{\Gamma}(p) = M_0 \Gamma_{bg}(p) + \sum_{k=1}^{K} M_k(p)\Gamma_k(p) \qquad (2)$$

where $\Gamma_{bg}(p)$ is prediction of background motion and $\Gamma_k(p)$ is prediction of target motion by K TPS transformations. After obtaining motion flow $\tilde{\Gamma}$ from $S$ to $I_t$, we use $\tilde{\Gamma}$ to deform $S$ and feature map $f_S$ of $S$, and then input them into the image generation network to obtain interpolated image $I_t$ at time $t$:

$$I_t = G(S^w, f_S^w) \qquad (3)$$

where $G(\cdot)$ Denote as image generation network, $S^w$ is image after deformation of $S$, and $f_S^w$ is feature after deformation of $f_S$. Similarly, we can obtain a series of interpolated images $\{I_t\}_{t=1}^{m}$, where $m$ is the number of interpolations.

**Motion transfer based on optimal interpolated image.** Traditional motion transfer methods usually directly model

motion from $S$ to $D$, but they often cannot obtain satisfactory results in the case of large displacement motion. Therefore, we propose a motion transfer method based on optimally interpolated images. Because keypoints of the interpolated image are obtained through keypoints of the source image and keypoints of the driving image, it makes the pose of the interpolated image be in the state between the source and driving images. Then, compared with large displacement motion estimation $\Gamma_{S->D}$ from $S$ to $D$, small displacement motion estimation $\Gamma_{I_t->D}$ from $I_t$ to $D$ will predict more accurately.

Then we use $I_t$ image to replace $S$ for motion transfer after obtaining interpolated image It. However, the quality of $I_t$ image will seriously affect the motion transfer's performance. Therefore, we design a selector for optimal interpolated image, which selects an image from a series of generated interpolated images that simultaneously satisfies the following requirements: 1) the identity information of the interpolated image should be intact; 2) The motion of the interpolated image to driving image should be as small as possible. The selector is set by the following:

$$t_{opt} = \arg\min_{t}\left(\frac{\|\Lambda(I_t) - \Lambda(S)\|}{\sum_{i=1}^{m}\|\Lambda(I_i) - \Lambda(S)\|} + \frac{\|Kp_I^t - Kp_D\|}{\sum_{i=1}^{m}\|Kp_I^i - Kp_D\|}\right) \qquad (4)$$

where $t_{opt}$ represents the index of optimal interpolated image, and $\Lambda(\cdot)$ represents appearance features of image extracted by pre-trained Vision Transformer (ViT) model, the introduction of ViT is shown in subsection 3.4. We calculate the appearance loss of all interpolated images and source images, and if the appearance information of the interpolated image is kept more complete, then the selection weight is larger. Moreover, distance loss from keypoints of all interpolated images to keypoints of the driving image is calculated. if the motion from the interpolated image to the driving image is a small displacement, then the selection weight is larger. The appearance loss and keypoint distance loss are normalized, and the optimal interpolated image is selected from interpolated images with the minimum sum of the loss, that is, the interpolated image with appearance information intact and small displacement from the driving image. After obtaining an optimal interpolated image, we use motion transfer from the optimal interpolated image to the driving image (Interpolation to Driving) to replace the traditional motion transfer (Source to Driving). Compared with the traditional methods, the pose of the transferred images obtained by the proposed method is more accurate. To learn high-quality optimal interpolated images, we propose a bidirectional training strategy.

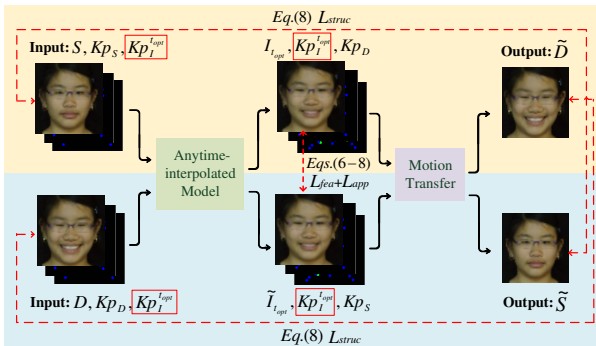

*Figure 2.* Bidirectional training strategy. It is worth noting that the interpolation module and the motion transfer module in both pipelines use the same optimal interpolation keypoints, as shown in red boxes.

### 3.3. Bidirectional Training Strategy

As shown in Figure 1, the pipeline is designed to generate an optimal interpolated image to decompose large displacement motion into small displacement motion. We can provide supervision at the end of the pipeline by extracting two frames from the video as input. However, only such supervision cannot generate satisfactory optimal interpolated images. Without monitoring the intermediate results, all subnets are treated as a whole to complete the motion migration task. In our framework, if there is no paired data to supervise, we should design some constraints to guarantee that the optimal interpolated image is meaningful. Otherwise, as long as the downstream motion transfer model can interpret the interpolated image, then the generator of the interpolation model can generate arbitrary interpolation results. If an effective constraint is found, the solution space can be reduced and a meaningful solution with the properties emphasized by the constraint can be obtained. To ensure the effectiveness of optimal interpolated images, we designed a bidirectional training strategy that does not require paired data.

As shown in Figure 2, the bidirectional training strategy consists of two pipelines: the upper pipeline is the motion transfer from $S$ to $D$, and the lower pipeline is the motion transfer from $D$ to $S$. In this process, the two pipelines generate optimal interpolation images $I_{t_{opt}}$ and $\tilde{I}_{t_{opt}}$ respectively by giving the same optimal interpolation keypoints $Kp_I^{t_{opt}}$. In theory, given the same interpolation keypoints, source and driving images with the same identity can often obtain the same interpolation image by interpolation model. Therefore, we add constraints to the optimal interpolated images $< I_{t_{opt}}, \tilde{I}_{t_{opt}} >$ such that they are consistent.

Our experiments demonstrated that the image generation network is crucial. Within our framework, both the interpo-

lation and motion transfer models utilize the shared-weight image generation network. If the image generation network in the motion transfer model exhibits strong generative capabilities, it can significantly enhance the quality of the optimal interpolated image generated.

### 3.4. Vision Transformers

In the motion transfer task, we aim to keep the source image's identity information and learn the driving image's pose. To make the generated image meet this requirement, we use pre-trained Vision Transformers(ViT) model(Caron et al., 2021; Tumanyan et al., 2022) to represent appearance information and structure information of image in the ViT feature space. With respect to appearance, we employ [CLS] to indicate a global image representation aimed at capturing both the informational content and stylistic attributes of global appearance. Regarding structure, the utilization of depth space features extracted by ViT, along with their self-similarity as structural representation, not only facilitates a robust representation of local texture but also ensures the preservation of the spatial layout and shape of the object and its surroundings.

In the bidirectional training strategy, we make use of the ability of ViT to extract appearance representation and structure representation and add appearance consistency and structure consistency constraints to generate optimal interpolated images ($I_{t_{opt}}$ and $\tilde{I}_{t_{opt}}$) to reduce their solution space. In addition, structure consistency constraint is added to the generated target image and driving image to promote the network and generate a better quality target image.

### 3.5. Loss Functions

**Reconstruction loss $L_{rec}$ :** To make the generated images $\tilde{D}$ and $\tilde{S}$ more realistic, we use the pre-trained VGG-19 network(Johnson et al., 2016) to calculate the perceptual loss of multi-resolution images:

$$
\begin{aligned}
L_{rec} = \sum_j \sum_i \left| V_i(D_j) - V_i(\tilde{D}_j) \right| \\
+ \sum_j \sum_i \left| V_i(S_j) - V_i(\tilde{S}_j) \right|
\end{aligned}
\tag{5}
$$

where $V_i$ is the $i$th layer of the pre-trained VGG-19 network, and $j$ indicates that the image has been downsampled $j$ times.

**Feature consistency loss $L_{fea}$ :** The encoder in Figure 1 is used to extract multi-scale features of optimal interpolated images ($I_{t_{opt}}$ and $\tilde{I}_{t_{opt}}$) in the bidirectional training process. The $L_{fea}$ loss encourages optimal interpolated images to be consistent. The loss is expressed as:

$$
L_{fea} = \sum_i \left| E_i(I_{t_{opt}}) - E_i(\tilde{I}_{t_{opt}}) \right|
\tag{6}
$$

| | Tai-Chi-HD | | | Fashion | | | UvA-Nemo | | | TedTalks | | |
|---|---|---|---|---|---|---|---|---|---|---|---|---|
| | $L_1$ | (AKD, MKR) | AED | $L_1$ | (AKD, MKR) | AED | $L_1$ | AKD | AED | $L_1$ | (AKD, MKR) | AED |
| X2Face | 0.080 | (17.65, 0.109) | 0.270 | - | - | - | 0.031 | 3.539 | 0.221 | - | - | - |
| FOMM | 0.057 | (6.65, 0.036) | 0.172 | 0.013 | (1.131, 0.006) | 0.059 | 0.021 | 1.408 | 0.067 | 0.033 | (7.07, 0.014) | 0.163 |
| MRAA | 0.048 | (5.41, 0.025) | 0.149 | - | - | - | 0.017 | 1.323 | 0.060 | 0.026 | (3.75, 0.007) | 0.114 |
| DAM | 0.044 | (4.79, 0.021) | 0.146 | **0.011** | (1.041, **0.004**) | **0.054** | - | - | - | - | - | - |
| MTIA | 0.045 | (4.67, 0.021) | 0.148 | - | - | - | - | - | - | 0.026 | (3.46, 0.007) | 0.113 |
| TPSMM | 0.045 | (4.57, 0.018) | 0.151 | **0.011** | (0.845, 0.005) | 0.056 | 0.011 | 1.177 | **0.050** | 0.027 | (3.39, 0.007) | 0.124 |
| CPABMM | **0.041** | (4.61, 0.021) | **0.117** | - | - | - | - | - | - | **0.022** | (3.21, 0.008) | **0.085** |
| Ours | 0.047 | (4.21, 0.014) | 0.157 | **0.011** | (0.800, **0.004**) | 0.056 | **0.010** | 1.155 | 0.051 | 0.028 | **(3.15, 0.005)** | 0.136 |
| Ours-V2 | 0.046 | **(3.67, 0.013)** | 0.149 | **0.011** | **(0.771, 0.004)** | 0.058 | **0.010** | **0.853** | **0.050** | - | - | - |

*Table 1.* Quantitative comparison of video reconstruction task on four different datasets.

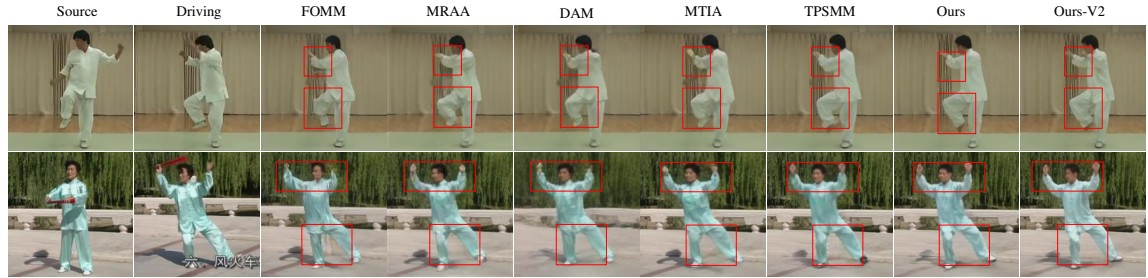

*Figure 3.* Some bad comparison methods cases on the Tai-Chi-HD dataset, while our method shows high quality on video reconstruction task.

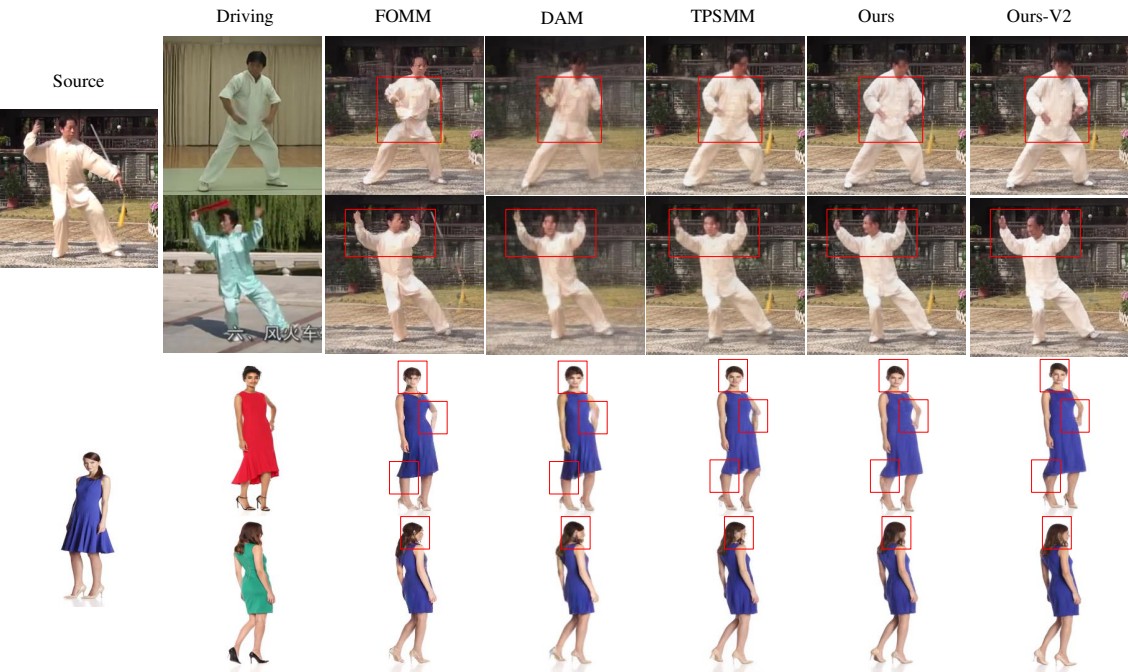

*Figure 4.* Qualitative comparison of image animation task on Tai-Chi-HD (above) and Fashion (below).

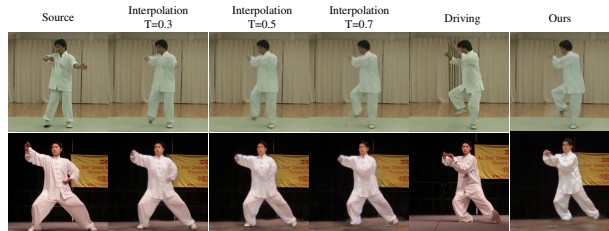

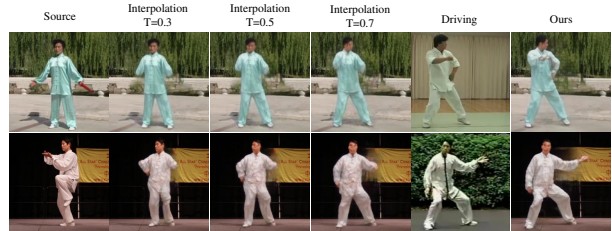

*Figure 5.* Interpolation results generated at different moments under the same identity.

*Figure 6.* Interpolation results generated at different moments under the different identity.

where $E_i$ represents layer $i$th of encoder.

**Appearance consistency loss $L_{app}$ :** The pre-trained ViT model(Caron et al., 2021; Tumanyan et al., 2022) is used to extract the appearance features of the optimal interpolated images $I_{t_{opt}}$ and $\tilde{I}_{t_{opt}}$ in bidirectional training process respectively, denoted as $\Lambda(I_{t_{opt}})$ and $\Lambda(\tilde{I}_{t_{opt}})$, where $\Lambda(\cdot)$ represents appearance features of images extracted by the ViT model. The appearance of the optimal interpolated image generated by $L_{app}$ loss is encouraged to be consistent, and loss is expressed as:

$$L_{app} = \left\| \Lambda(I_{t_{opt}}) - \Lambda(\tilde{I}_{t_{opt}}) \right\|_2 \qquad (7)$$

**Structural consistency loss $L_{struc}$ :** The self-similarity matrix of $I_{t_{opt}}, \tilde{I}_{t_{opt}}, S, \tilde{S}, D$, and $\tilde{D}$ is extracted using the pre-trained ViT model(Caron et al., 2021; Tumanyan et al., 2022), denoated as $\Phi(I_{t_{opt}}), \Phi(\tilde{I}_{t_{opt}}), \Phi(S), \Phi(\tilde{S}), \Phi(D)$, and $\Phi(\tilde{D})$ respectively, where $\Phi(\cdot)$ represents structural information of image extracted by ViT model. The structural consistency loss is calculated as:

$$L_{struc} = \left\| \Phi(I_{t_{opt}}) - \Phi(\tilde{I}_{t_{opt}}) \right\|_F + \left\| \Phi(S) - \Phi(\tilde{S}) \right\|_F + \left\| \Phi(D) - \Phi(\tilde{D}) \right\|_F \qquad (8)$$

In general, the total training loss function is:

$$L_{loss} = \lambda_r L_{rec} + \lambda_f L_{fea} + \lambda_a L_{app} + \lambda_s L_{struc} \qquad (9)$$

where $\lambda_r, \lambda_f, \lambda_a$, and $\lambda_s$ are hyperparameters.

# 4. Experiments

## 4.1. Benchmarks

**Datasets:** We trained on multiple types of datasets, including faces and human bodies. The datasets are as follows: UvA-Nemo(Dibeklioglu et al., 2012; 2015), Fashion(Zablotskaia et al., 2019), Tai-Chi-HD(Siarohin et al., 2019b), and TedTalks(Siarohin et al., 2021)

**Metrics:** Same as previous work, in the video reconstruction task with the same identity, the first frame $D_1$ of video is used as a source image to reconstruct $\{D_t\}_{t=1}^n$. The quantitative indicators (Siarohin et al., 2019b) used are L1, Average Keypoint Distance (AKD), Missing Keypoint Rate (MKR), and Average Euclidean Distance (AED).

**Implementation Details:** The method is implemented on PyTorch(Paszke et al., 2019), all experiments are conducted on an NVIDIA 4090 GPU with a resolution of $256 \times 256$ for all datasets and 100 epochs of training. We use Adam optimizer(Kingma & Ba, 2015) to update our model and set the learning rate to 0.0001, which dropped by a factor of 10 at the end of the 70th epoch and the 90th epoch. We set the training hyperparameters to: $\lambda_r = 10$, $\lambda_f = 10$, $\lambda_a = 10$, and $\lambda_s = 10$. The interpolated images are generated at $\{0.3, 0.5, 0.7\}$ respectively, and then optimal interpolated images that meet the conditions are selected.

## 4.2. Comparison

We compare the proposed method with the state-of-the-art unsupervised motion transfer methods (X2Face(Wiles et al., 2018), FOMM, MRAA, DAM(Tao et al., 2022b), MTIA(Tao et al., 2022a), TPSMM, and CPABMM) for video reconstruction and image animation tasks.

**Video reconstruction:** The quantitative results of Video reconstruction are shown in Table 1. Our approach achieved significant improvements in motion-related metrics on four datasets. Compared with TPSMM, on the Tai-Chi-HD dataset, AKD and MKR improved by 7.88% and 22.2%, respectively, and on the Fashion dataset, AKD and MKR increased by 5.3% and 20%, respectively. on the TedTalks dataset, Compared with CPABMM, AKD and MKR increased by 1.87% and 37.5%. It shows that our model can obtain a pose closer to the real image.

As shown in Figure 3, we show qualitative results of video reconstruction, which prove that the interpolation method between source and driving images can obtain more accurate estimation of large displacement motion. However,

| | L1 | (AKD, MKR) | AED |
|---|---|---|---|
| TPSMM | **0.045** | (4.57,0.018) | **0.151** |
| +TPSMM, INT (Model1) | 0.053 | (5.56,0.026) | 0.169 |
| +Model1, BT (Model2) | 0.048 | (4.31,0.016) | 0.158 |
| +Model2, $L_{app}$, $L_{struc}$ (Ours) | 0.047 | **(4.21,0.014)** | 0.157 |

*Table 2.* Ablating the key components of the proposed method. INT is denoted as adding an unsupervised anytime interpolation method, and BT is denoted as adding a bidirectional training strategy.

the interpolation method also causes a loss of appearance information. As shown in Figures 5 and 6, we show the interpolation at different moments and final generated results. Compared with the source image, the generated interpolated image already has the problem of blurring, which leads to the appearance effect of the target image generated by the downstream task is not satisfactory. In addition, for the dataset with background (Tai-Chi-HD), the interpolation method causes occlusion region to expand, as shown by the background in the second row of Figure 6. This is also why the $L_1$ and AED metrics in Table 1 are slightly lower than the existing methods.

**Image Animation:** Figure 4 shows the comparison results of motion transfer between the proposed method and comparison methods under different identities. Experimental results show that the proposed method has better motion transfer performance for large displacement motion of the human body (Tai-Chi-HD and Fashion). However, the ability to maintain the details of images, such as clothing and face, is slightly worse. As shown in the second row of Figures 5 and 6, the clothing color of the interpolated image generated has a color difference with the source image, resulting in a large difference between the appearance of the image generated by downstream motion transfer and the appearance of the source image.

### 4.3. Ablations

In this section, we conduct comprehensive ablation experiments on the Tai-Chi-HD dataset to systematically evaluate the contribution of key components to the final performance of the method. We successively added our proposed components; the results are shown in Table 2. The second row of Table 2 represents the results of the TPSMM method, Model1 represents the addition of an unsupervised anytime interpolation method to the TPSMM method, Model2 is the addition of a bidirectional training strategy based on Model1, and Full model is our proposed model.

For the experimental results of Model1, we find that adding an unsupervised anytime interpolation method to the TPSMM method not only does not improve the model's

performance but also decreases the model's performance. This is because the interpolation method results in poor generation of interpolated images without constraint information, which affects the downstream motion transfer task. For Model2, when we add the bidirectional training strategy, the generation of interpolated images is appropriately constrained, improving motion transfer performance. For the Full model (Ours), we add appearance consistency loss and structure consistency loss so that the quality of the generated images is improved and optimal in the motion-related metrics.

### 4.4. Method Extension

The proposed method is extended to multi-view tasks according to the optimal interpolation method. Specifically, source image and optimally interpolated images are used to constitute multi-view data. The source image can provide better appearance information, and the motion displacement between the optimal interpolated image and the driving image is smaller, which can learn the pose information more accurately. Therefore, in the motion transfer network, we fuse the deformation features of the source image and the optimally interpolated image, and the experimental results are shown in Ours-V2 of Table 1. Compared with the proposed single-view motion transfer method (Ours), Ours-V2 achieves significant improvements on most datasets. Among them, the motion-related metric AKD is increased by 12.8%, 3.6% and 26.1% on Tai-Chi-HD, Fashion and UvA-Nemo, respectively. As shown in Figures 3 and 4, the appearance of the generated images is significantly improved.

## 5. Conclusion

To solve large displacement motion, we propose a large displacement motion transfer model based on unsupervised anytime interpolation. Compared with the previous method of directly predicting the motion from the source image to the driving image, the proposed method interpolates the source image and driving image at any time, decomposes the original large displacement motion into multiple small displacement motions, and improves the accuracy of motion estimation. To achieve unsupervised interpolation, we propose a bidirectional training strategy, which adds constraints to the intermediate interpolated images to narrow the generation range of interpolated images. The pre-trained Vit model adds appearance and structure consistency to the images so that the generated images have better quality. Experiments show that our method achieves optimal performance on motion-related metrics compared to other state-of-art methods.

## Acknowledgements

This work was partly supported by the Key Research and Development Plan of Zhejiang: No.2021C03131; National Natural Science Fund of China No.62471170 & No.61871170; Zhejiang Province Fund: LQ21F020005.

## Impact Statement

This paper presents work whose goal is to advance the field of Machine Learning. There are many potential societal consequences of our work, none which we feel must be specifically highlighted here.

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
