# OpenReview forum: "Large Displacement Motion Transfer with Unsupervised Anytime Interpolation"
_ICML.cc/2025/Conference — ICML 2025 poster_

### Official Review · Reviewer_swRi · 2025-03-12

**Overall Recommendation:** 3

**Summary:**

This paper presents an anytime interpolation framework for flexible and accurate motion-driven frame generation. Specifically, during training, the model searches for an optimal intermediate time step that produces the highest-quality interpolated frame for training. To ensure valid motion transfer, the authors design an unsupervised bidirectional training strategy, effectively preserving appearance and structural consistency in the generated frames.

**Claims And Evidence:**

The majority of the content is clear, but some detailed design aspects need further justification.

**Essential References Not Discussed:**

NA

**Ethical Review Concerns:**

The visual examples and the training/evaluation datasets mayraise concerns regarding privacy and security.

**Ethical Review Flag:**

Flag this paper for an ethics review.

**Ethics Expertise Needed:**

["Privacy and Security"]

**Experimental Designs Or Analyses:**

Yes, the authors provide several experimental details to validate the algorithm, but some ablation analyses are missing.

**Methods And Evaluation Criteria:**

The evaluation criteria appear to be appropriate and sufficient.

**Other Comments Or Suggestions:**

See the ``Other Strengths And Weaknesses"

**Other Strengths And Weaknesses:**

Strengths:

The motion decomposition strategy alleviates the challenge of directly transferring large displacement motions.
To enhance supervision, the authors propose an optimal interpolation selector, which identifies the interpolated frame with the best identity consistency and minimal warping distance.
The method incorporates appearance and structural consistency constraints, improving the overall motion transfer quality.

Weaknesses & Suggested Revisions

The use of L1/L2-based reconstruction loss may lead to blurry interpolated results. Consider integrating GAN-based refinement or a perceptual loss to improve sharpness.
The design of the interpolation selector lacks sufficient justification. A more detailed discussion on the impact of different weighting strategies on final performance would strengthen the argument.
The model requires pre-generating multiple interpolated frames and selecting the best one, which may introduce significant computational overhead. It would be beneficial to provide a comparison of inference/training costs with other baseline methods.
The paper should discuss and compare conditional diffusion-based approaches, which are gaining popularity in motion transfer. Highlighting the advantages of the proposed method over such alternatives would further clarify its contributions.

**Questions For Authors:**

NA

**Relation To Broader Scientific Literature:**

Provide insights into unsupervised large motion transfer, such as the search for optimal interpolated time steps and constraints on appearance and structural consistency for effective decomposition.

**Theoretical Claims:**

No theoretical claims

---

> ### Author Rebuttal · Authors · 2025-04-01
>
> Thank you for the valuable comments.
>
> Q1: The authors provide several experimental details to validate the algorithm, but some ablation analyses are missing.
>
> A1: Thank you for your comments. The Reviewer 3 LLC asked the same question in Q3.
> In the ablation experiment, we mainly validate three modules, the first is the interpolation method. We added the interpolation method on the baseline (TPSMM), as shown in Model 1 in Table 2. In the unsupervised case, the lack of constraints on the interpolated image makes the generated interpolated image very poor, which seriously affects the quality of the downstream target image generation. To solve this problem, we add a second module, the two-way training strategy, as shown in Model 2 in Table 2. By adding consistency constraints to the intermediate generated interpolated image pairs, the generation range of the interpolated images is narrowed down, significantly improving the generation quality of the target images. In order to improve the generation quality of the target image, we added a third module to add appearance consistency loss and structural consistency loss to the image using the pre-trained Vit model, as shown in “Ours” in Table 2, which is our complete model.
>
>
> Q2: The use of L1/L2-based reconstruction loss may lead to blurry interpolated results. Consider integrating GAN-based refinement or a perceptual loss to improve sharpness.
>
> A2: Thank you for your comments. In the unsupervised scenario, we aimed to improve the quality of the interpolated images through bidirectional training by incorporating additional loss functions to constrain their generation. Initially, we used L1 constraints on the generated pairs of interpolated images; however, our experiments showed that these L1 constraints were too strict. As a result, minimizing the loss on these image pairs often led to the generation of all-white images. We applied L1 loss to their multiscale features to address this issue, as demonstrated in Equation (6).
>
> To further enhance image quality, we utilized a pre-trained Vision Transformer (ViT) model to introduce appearance consistency loss and structure consistency loss to the interpolated image pairs, as outlined in Equations (7) and (8). Additionally, we attempted to refine the images using a super segmentation network, but the improvements were minimal.
>
> Q3: The design of the interpolation selector lacks sufficient justification. A more detailed discussion on the impact of different weighting strategies on final performance would strengthen the argument.
>
> A3: The interpolation network generates multiple interpolated images that progressively adopt the poses of the driving image as time (t) increases from 0 to 1. However, we observe that as the pose gets closer to that of the driving image, the visual quality of the interpolated images tends to deteriorate. To address this issue, we have designed an interpolation selector aimed at identifying the optimal interpolated image from the set of generated images. This optimal image should not only have a pose that closely matches the driving image but also maintain visual consistency with the source image. In motion migration, preserving appearance is just as crucial as learning the pose; therefore, we assign equal weights in Equation (4). In our future studies, we plan to explore how different weighting strategies might affect overall performance.
>
> Q4: The model requires pre-generating multiple interpolated frames and selecting the best one, which may introduce significant computational overhead.
>
> A4: Thank you for your valuable comments, multiple interpolated frames and selecting the optimal interpolated frame impose a significant computational overhead on the model. Since we are unsupervised interpolation method, there is no guarantee that the generated interpolated images can satisfy the subsequent downstream tasks without labeling. In my future research, I will improve the proposed model as a way to reduce the computational overhead.
>
> Q5: The paper should discuss and compare conditional diffusion-based approaches, which are gaining popularity in motion transfer. Highlighting the advantages of the proposed method over such alternatives would further clarify its contributions.
>
> A5: Thank you for your valuable comments. In recent years, diffusion-based motion migration methods have shown satisfactory performance. However, these methods are supervised and often rely on a pre-trained model to extract a priori conditions for the target image, such as human body key points. In the case of the Taichi dataset, accurately extracting human body poses can be challenging due to the low resolution of the images. Additionally, the diffusion process requires extensive computational resources, which may limit the model's performance in environments with restricted resources.

---

### Official Review · Reviewer_cLLC · 2025-03-14

**Overall Recommendation:** 3

**Summary:**

The proposed method advances unsupervised motion transfer by addressing the challenge of large displacement motions through interpolation and strategic training. While it excels in pose accuracy, it faces minor challenges in maintaining appearance details, particularly in complex scenarios. This work provides a robust framework for applications in image animation, with potential for further refinement in appearance preservation.

**Claims And Evidence:**

The linear motion assumption is too strict and not empirically validated.

The lack of comparison with recent works makes it uncertain if the method is truly state-of-the-art.

The ablation study does not clearly prove the necessity of each module.

**Essential References Not Discussed:**

No

**Experimental Designs Or Analyses:**

Comparisons should include recent methods (2023-2024) to validate performance.

Ablation studies need more detailed analysis to confirm the contribution of each module.

**Methods And Evaluation Criteria:**

The method is reasonable but assumes linear motion, which may not hold for complex cases. The datasets are appropriate, but lack comparisons with newer methods.

**Other Comments Or Suggestions:**

No

**Other Strengths And Weaknesses:**

Strength:

1.	The author proposed to decompose the traditional motion transfer pipeline into two steps to solve the difficulty when there is large displacement of motion between S and D.

2.	A ViT is introduced to ensure the consistency of appearance and motion.

Weakness:

1.	The assumption that motion between S and D of each keypoint is linear seems to be strict, especially in cases such as non-linear human actions in Tai-Chi-HD.

2.	In quantitative comparison, the newest work is from 2022. More recent works are better be included.

3.	In ablation study, the effectiveness of the proposed module is not clearly proved according to Table.2.

4.  Some related works [1-2] are discussed and compared.

[1]. Structure-aware motion transfer with deformable anchor model. CVPR 2022.

[2]. Motion Transformer for Unsupervised Image Animation. ECCV 2022.

**Questions For Authors:**

No

**Relation To Broader Scientific Literature:**

The paper extends prior motion transfer work with a two-step strategy and ViT-based consistency enforcement, but its linear motion assumption, outdated baselines, and missing perceptual metrics limit its broader impact.

**Theoretical Claims:**

Since the paper primarily relies on empirical validation, no formal proof of correctness check is required.

---

> ### Author Rebuttal · Authors · 2025-04-01
>
> Thank you for the comments and suggestions you gave. We have incorporated your feedback into the paper.
>
> Q1: The method is reasonable but assumes linear motion, which may not hold for complex cases.
>
> A1: In the interpolation method, we assume that the local keypoints' motion is linear to obtain keypoints that do not exist between the source and driver images. The corresponding interpolated images are then generated from these interpolated keypoints. The pose of these interpolated images is between the source and driver images. The motion from the interpolated image to the driver image is smaller than the magnitude from the source image to the driver image. And for modeling the motion, we use the nonlinear transform (TPS) from the literature [1] to approximate the motion.
>
> [1] Zhao J, Zhang H. Thin-plate spline motion model for image animation[C]//Proceedings of the IEEE/CVF Conference on Computer Vision and Pattern Recognition. 2022: 3657-3666.
>
> Q2: The datasets are appropriate, but lack comparisons with newer methods.
>
> A2: We have added the 3 most recent comparison methods (DAM [2], MTIA [3], and CPABMM [4]) as shown in Table 1 in A2 of Reviewer 1 LsVD. Relevant qualitative results have also been added, as shown in the link (https://github.com/ICML2025Anonymity/Anonymity), which complies with the double-blind policy.
>
> [2] Tao J, Wang B, Xu B, et al. Structure-aware motion transfer with deformable anchor model[C]//Proceedings of the IEEE/CVF conference on computer vision and pattern recognition. 2022: 3637-3646.
>
> [3] Tao J, Wang B, Ge T, et al. Motion transformer for unsupervised image animation[C]//European conference on computer vision. Cham: Springer Nature Switzerland, 2022: 702-719.
>
> [4] Wang H, Liu F, Zhou Q, et al. Continuous piecewise-affine based motion model for image animation[C]//Proceedings of the AAAI Conference on Artificial Intelligence. 2024, 38(6): 5427-5435.
>
> Q3: In ablation study, the effectiveness of the proposed module is not clearly proved according to Table.2.
>
> A3: In the ablation experiment, we mainly validate three modules, the first is the interpolation method. We added the interpolation method on the baseline (TPSMM), as shown in Model 1 in Table 2. In the unsupervised case, the lack of constraints on the interpolated image makes the generated interpolated image very poor, which seriously affects the quality of the downstream target image generation. To solve this problem, we add a second module, the two-way training strategy, as shown in Model 2 in Table 2. By adding consistency constraints to the intermediate generated interpolated image pairs, the generation range of the interpolated images is narrowed down, significantly improving the generation quality of the target images. In order to improve the generation quality of the target image, we added a third module to add appearance consistency loss and structural consistency loss to the image using the pre-trained Vit model, as shown in “Ours” in Table 2, which is our complete model.

---

### Official Review · Reviewer_usGM · 2025-03-14

**Overall Recommendation:** 2

**Summary:**

This paper propose a unsupervised motion transfer algorithm that can transfer pose in the driving video to the object of the source image so that the source image can copy the movement of the driving video. To be exact, the method decompose complex large displacement motion into many small displacement motions, and improve the accuracy of motion estimation. A bi-directional training strategy is used to constrain the intermediate interpolated images.

**Claims And Evidence:**

The experimental results are not convincing, such as when comparing with state-of-the-art qualitatively, it only shows one state-of-the-art method (TPSMM) in figure 3, and figure 5. It did not compare with X2Face, FOMM, MRAA, I wonder why?

**Essential References Not Discussed:**

In the experiment part, the paper compares with a few state-of-the-art algorithms including X2Face, but X2Face is not mentioned/referred in the reference. It's difficult to understand what is exactly X2Face.

**Experimental Designs Or Analyses:**

When comparing with state-of-the-art, it's essential to compare with the most up to date and relevant methods. X2Face is not referred, so I don't know which paper it is referring to. FOMM was published in 2019. MRAA was published in 2021, TPSMM was published in 2022. Why not compare with any more recent papers such as
Wang, H., Liu, F., Zhou, Q., Yi, R., Tan, X., and Ma, L. Continuous piecewise-affine based motion model for image animation. In In Proceedings of the Thirty-Eighth AAAI Conference on Artificial Intelligence, AAAI, pp. 5427–5435, 2024.
From this paper's experiment results, it seems that this paper has much better performance.


Qualitative comparison seems not convincing, as shown in Figure 3-6, the proposed method's results look quite blurry, and images are of low resolution. Also since the paper claims to deal with large displacement motion transfer, I find that in the figures, a lot of examples have small displacement motion transfer, such as Figure 3 (first row), Figure 4 (second and third rows), Figure 6 (second the third rows). Why not showing more results that have large motion displacement to highlight the paper's main claim?

When conducting the quantitative experiment, the proposed methods leave out a few key evaluation datasets, such as "TED-talks" , "VoxCeleb",  "MGif". I wonder why this paper did not compare with state-of-the-art methods on these datasets.

**Methods And Evaluation Criteria:**

Evaluation follow criteria that used by state-of-the-art. But the experiment lack a few key datasets.

**Other Comments Or Suggestions:**

There seems to be a grammar issue in section 3.2 "Inspired by it, a keypoint-based anytime interpolation method", is this an un-finished sentence?

**Other Strengths And Weaknesses:**

The main strength is that the paper claims to deal with a challenging tasks of large displacement motion transfer. The weakness is the lack of experimental results to support the claim,

**Questions For Authors:**

My question are mainly related to the experimental design and comparison with more recent/relevant papers.

**Relation To Broader Scientific Literature:**

The proposed method address a small research area of motion transfer.

**Theoretical Claims:**

looks correct.

---

> ### Author Rebuttal · Authors · 2025-04-01
>
> Thank you for your valuabe comments. We have incorporated your feedback into the maniscript. We believe it will help strengthen the work and present it better.
>
> Q1: The experimental results are not convincing, such as when comparing with state-of-the-art qualitatively, it only shows one state-of-the-art method (TPSMM) in figure 3, and figure 5. It did not compare with X2Face, FOMM, MRAA, I wonder why?
>
> A1: Thank you for the valuable feedback. We have provided more comparative results in the following link: https://github.com/ICML2025Anonymity/Anonymity. This is fully compliant with the double-blind policy. In response to the reviewer's question, our decision to compare the proposed method primarily with TPSMM is based on existing literature, which shows that TPSMM outperforms X2Face, FOMM, and MRAA. Therefore, we have chosen to validate the superiority of our method by comparing it only to TPSMM.
>
> Q2: X2Face is not referred. Why not compare with any more recent papers such as Wang, H., et al. Continuous piecewise-affine based motion model for image animation.
>
> A2: Thank you for your valuable comments, and the manuscript has been revised and cited more recent literature papers such as Wang, H., et al. Also,  we added comparison results with CPABMM, as the answer to Reviewer 1 LsVD's A3.
>
> Q3: Qualitative comparison seems not convincing, as shown in Figure 3-6, the proposed method's results look quite blurry, and images are of low resolution. Also since the paper claims to deal with large displacement motion transfer, I find that in the figures, a lot of examples have small displacement motion transfer, such as Figure 3 (first row), Figure 4 (second and third rows), Figure 6 (second the third rows). Why not showing more results that have large motion displacement to highlight the paper's main claim?
>
> A3: Thanks to the reviewers' insightful feedback, we show more results with large motion displacements to highlight the paper's main points in this link (https://github.com/ICML2025Anonymity/Anonymity). We modify the presentation of Figures 3-6, and the process of motion transfer is all large displacement motion, such as https://github.com/ICML2025Anonymity/Anonymity/tree/main/Figues%203-6.
>
> Q4: Why are datasets such as "TED-talks", "VoxCeleb", "MGif" not used when quantifying experiments?
>
> A4: Thank you for your comments, and Reviewer 1 LsVD also raised a similar question in Q1, and we addressed it in our response A1 to Reviewer LsVD. The unsupervised optimal interpolation method proposed in this paper aims to address the large motion problem in motion migration. To validate the effectiveness of this method, we selected two datasets—TaiChiHD and Fashion—that exhibit significant motion amplitudes.
>
> While the face dataset typically demonstrates a smaller range of motion compared to larger human datasets, we included the UvA-Nemo dataset to assess the method’s performance in scenarios with small motion amplitudes. Although VoxCeleb is another face dataset, its size—approximately 308 GB—restricts its usability.
>
> The Ted Talks dataset has a resolution of 384×384, which is higher than the resolution of the datasets we used. However, the motion of the human body in the Ted Talks videos is minimal, as illustrated in https://github.com/ICML2025Anonymity/Anonymity/tree/main/TedTalks, by the double-blind policy. Due to the constraints of the rebuttal date, we only conducted experiments using the Ted Talks dataset.
>
> Regarding the MGif cartoon animal dataset, we were unable to evaluate motion-related metrics like AKD, which means that the dataset is not appropriate for validating our claim that our method enhances postural accuracy.
>
> Q5: The main strength is that the paper claims to deal with a challenging tasks of large displacement motion transfer. The weakness is the lack of experimental results to support the claim.
>
> A5: In this paper, we deal with the problem of large displacements between source and driver images by interpolating between them. As in Fig. 4 and Fig. 6, we show the interpolated image under the same identity and the interpolated image under different identities, respectively. Visually, it is observed that the movements of the interpolated image are closer to those of the driving image than those of the source image, which significantly alleviates the large displacement problem in the motion migration task. More experimental results will be shown in the following link (https://github.com/ICML2025Anonymity/Anonymity).
>
> Q6: There seems to be a grammar issue in section 3.2 "Inspired by it, a keypoint-based anytime interpolation method", is this an un-finished sentence?
>
> A6: Thank you for your careful review. The error has been revised: "Inspired by it, a keypoint-based anytime interpolation method is proposed".

---

### Official Review · Reviewer_LsVD · 2025-03-14

**Overall Recommendation:** 3

**Summary:**

This paper proposes a novel method for transferring large motion from a driving to a source image. The proposed method is to find a middle step which essentially adds non linearity to the motion transfer. The method generates a set of interpolated in between images based on key point transfers and selects the optimal interpolation point. Then transfers the motion from the interpolated image to the driving image, in effect takes a shorter final step. In order to constrain the optimal interpolations the method adopts a bidirectional training scheme where both the (source -> driving) and (driving -> source) is considered and the hypothesis is to have the same optimal interpolation point between the two directions.
The results show improvements in terms of keypoint accuracy.

**Claims And Evidence:**

The claim is that the interpolation helps with large motion transfer. Overall the accuracy of the keypoints are improved  but there are no ablations on large or excessively large motion that is not usually considered in other works. If any some of the typical datasets such as Ted-talks and voxceleb are missing.

**Essential References Not Discussed:**

none

**Experimental Designs Or Analyses:**

The experiments are missing results on TedTalks and VoxCeleb. The current evaluations show that the bidirectional training improves the accuracy of the keypoints significantly. There is not much discussion why the L1 and AED metrics are not on par with SoTA, specially considering CPABMM it seems that the quality of the generated images might be suffering from the interpolation.

**Methods And Evaluation Criteria:**

yes, the method makes sense for motion transfer and the evaluation criteria are well established.

**Other Comments Or Suggestions:**

TPSMM on taichiHD is usually 4.57 AKD, why is it higher here? It would be better to report the official AKD for TPSMM and add CPABMM to the table as well.

The paper is not in review format.

Page 6, 2nd paragraph, [21 - 22] seems to be a citation syntax error.

eq. 5: break so that it doesn't go over text width.

**Other Strengths And Weaknesses:**

The idea is novel and significantly improves the results in terms of keypoint accuracy.

The write up has many syntax and grammatical errors but it still reads fine.

The main weakness is not quantifying what a "large" motion is, selecting datasets that show "larger" motion than typical ones and reporting results on Ted Talks.

**Questions For Authors:**

what is the difference between features used in the appearance loss and structure loss? both of them seems to be extracted from the same network. are they from different layers? which layers?

**Relation To Broader Scientific Literature:**

The notion of using interpolation and incremental steps is already well established. But the idea of finding an optimal interpolation point and using bidirectional training to fix that point is novel and seems to be significantly improving the keypoint accuracy.

**Theoretical Claims:**

no theoretical claim.

---

> ### Author Rebuttal · Authors · 2025-04-01
>
> Thank you for the comments and suggestions.
>
> Q1: The experiments are missing results on TedTalks and VoxCeleb.
>
> A1: The unsupervised optimal interpolation method proposed in this paper aims to address the large motion problem in motion migration. To validate the effectiveness of this method, we selected two datasets—TaiChiHD and Fashion—that exhibit significant motion amplitudes.
>
> While the face dataset typically demonstrates a smaller range of motion compared to larger human datasets, we included the UvA-Nemo dataset to assess the method’s performance in scenarios with small motion amplitudes. Although VoxCeleb is another face dataset, its size—approximately 308 GB—restricts its usability.
>
> The Ted Talks dataset has a resolution of 384×384, which is higher than the resolution of the datasets we used. However, the motion of the human body in the Ted Talks videos is minimal, as illustrated in https://github.com/ICML2025Anonymity/Anonymity/tree/main/TedTalks, by the double-blind policy. Due to the constraints of the rebuttal date, we only conducted experiments using the Ted Talks dataset.
>
> Q2: There is not much discussion why the L1 and AED metrics are not on par with SoTA, the quality of the generated images might be suffering from the interpolation.
>
> A2: Although the interpolated image's quality significantly impacts the final target image's quality, it helps reduce the substantial motion between the source and driving images to a smaller motion between the interpolated and driving images, significantly improving the AKD metric.
>
> To address the issues of L1 and AED degradation, we have included enhancements to the model in the manuscript and added relevant content to subsection 4.4. Specifically, the method based on optimal interpolation provides better poses for the model, while the motion transfer technique from the source image to the driving image enhances the appearance. By combining the advantages of these two approaches, we achieved significant improvements demonstrated in Ours-V2 in Table 1, linked to https://github.com/ICML2025Anonymity/Anonymity/tree/main/Table%201.
>
> Q3: This paper does not compare with the recent paper CPABMM
>
> A3: Although CPABMM performed well in the motion migration task, the paper's authors did not provide pre-training checkpoints, and the training process is quite time-consuming. In our experimental setting, reproducing the original results was challenging. As a result, we utilized the official results. Thanks for your suggestion again. Since the rebuttal time limitation, we will try to reproduce the results when we have the time. On the other hand, our method still achieves optimal performance in the AKD metric compared to CPABMM on the TaiChiHD dataset.
>
> Q4: What is the difference between features used in the appearance loss and structure loss? Both of them seems to be extracted from the same network. Are they from different layers? Which layers?
>
> A4: Appearance loss involves comparing two images' [CLS] token. The [CLS] token serves as the global semantic descriptor in Vision Transformers (ViT), akin to the “full-image feature” found in Convolutional Neural Networks (CNNs). This token is derived from the last layer of ViT and emphasizes the overall semantics of the images.
>
> On the other hand, structural loss focuses on comparing the self-similarity of two images using the Key matrix from the 11th layer of ViT. The self-similarity matrix generated by the Keys at this deeper layer reveals the global correlation patterns among different image regions. As such, it can be utilized to assess the structural similarity between the two images.
>
> Q5: TPSMM on taichiHD is usually 4.57 AKD, why is it higher here? It would be better to report the official AKD for TPSMM and add CPABMM to the table.
>
> A5: For TPSMM and FOMM, the experimental results presented in Table 1 are those we reproduced under the existing experimental conditions. We have updated the results to reflect the official findings and included several recent methods, such as CPABMM, in the experiments, as shown in Table 1.
>
> Q6: Page 6, 2nd paragraph, [21 - 22] seems to be a citation syntax error. Eq. 5: break so that it doesn't go over text width.
>
> A6: The manuscript has been revised, and the errors have been corrected.

---

### Decision · Program_Chairs · 2025-05-01

**Decision:**

Accept (poster)

**Comment:**

This paper is about transferring large motion from a driving video to a source image.

The paper initially received three weak accept and one weak reject. The main strength lies in the novel bi-directional training strategy and optimal interpolation selector. The reviewers appreciate the novelty of these components and the improvement over the baselines. The main weaknesses are mainly on the lack of comparisons with more recent methods and missing results on the common datases like TedTalks and VoxCeleb.

In the authors' rebuttal, the authors provided additional experimental comparisons with three recent methods (DAM, MTIA, and CPABMM) in Table 1 in the provided link, as well as results on TedTalks. Overall, the three reviewers LsVD, cLLC, and swRi maintain their positive views about this paper while Reviewer usGM did not respond or acknowledge the rebuttal. The AC read the reviews and the rebuttal and found that the additional results sufficiently addressed the raised concerns. Consequently, the AC recommends to accept.